# Sensor-Location-Specific Joint Acquisition of Peripheral Artery Bioimpedance and Photoplethysmogram for Wearable Applications

**DOI:** 10.3390/s23167111

**Published:** 2023-08-11

**Authors:** Margus Metshein, Anar Abdullayev, Antoine Gautier, Benoit Larras, Antoine Frappe, Barry Cardiff, Paul Annus, Raul Land, Olev Märtens

**Affiliations:** 1Thomas Johann Seebeck Department of Electronics, Tallinn University of Technology, Ehitajate Tee 5, 19086 Tallinn, Estonia; 2University Lille, CNRS, Centrale Lille, Junia, University Polytechnique Hauts-de-France, UMR 8520-IEMN, F-59000 Lille, France; 3School of Electrical and Electronic Engineering, University College Dublin, D04V1W8 Dublin, Ireland

**Keywords:** cardiovascular system, convolutional neural networks, electrical bioimpedance, deep learning, photoplethysmography, non-invasive measurements, pulse wave, sensor fusion, small data machine learning, wearable devices

## Abstract

Background: Cardiovascular diseases (CVDs), being the culprit for one-third of deaths globally, constitute a challenge for biomedical instrumentation development, especially for early disease detection. Pulsating arterial blood flow, providing access to cardiac-related parameters, involves the whole body. Unobtrusive and continuous acquisition of electrical bioimpedance (EBI) and photoplethysmography (PPG) constitute important techniques for monitoring the peripheral arteries, requiring novel approaches and clever means. Methods: In this work, five peripheral arteries were selected for EBI and PPG signal acquisition. The acquisition sites were evaluated based on the signal morphological parameters. A small-data-based deep learning model, which increases the data by dividing them into cardiac periods, was proposed to evaluate the continuity of the signals. Results: The highest sensitivity of EBI was gained for the carotid artery (0.86%), three times higher than that for the next best, the posterior tibial artery (0.27%). The excitation signal parameters affect the measured EBI, confirming the suitability of classical 100 kHz frequency (average probability of 52.35%). The continuity evaluation of the EBI signals confirmed the advantage of the carotid artery (59.4%), while the posterior tibial artery (49.26%) surpasses the radial artery (48.17%). The PPG signal, conversely, commends the location of the posterior tibial artery (97.87%). Conclusions: The peripheral arteries are highly suitable for non-invasive EBI and PPG signal acquisition. The posterior tibial artery constitutes a candidate for the joint acquisition of EBI and PPG signals in sensor-fusion-based wearable devices—an important finding of this research.

## 1. Introduction

A crucial aspect that the early detection of cardiovascular diseases (CVDs) depends on is the existence of suitable monitoring devices and signal-processing methods. For unobtrusive monitoring, devices, desirably wearable, capable of processing the signals in a way that enables extracting indicative features (e.g., morphological) are required. While bedside patient monitors enable the application of standard configurations of measurement instrumentation (like the 12-lead electrocardiography (ECG), fingertip photoplethysmography (PPG), thoracic electrical bioimpedance (EBI)-based impedance cardiography (ICG) [1], etc.), the vastly emerging need for wearable devices is posing additional challenges. Such areas are blood pressure monitoring [2] and frailty-syndrome-related comorbidity-specific marker detection, being remarkably linked to the cardiovascular system [3].

A promising method for peripheral artery monitoring is the EBI, relying on the excitation of the object with a small value electrical current or voltage and measuring the response [1]. The method enables the detection of static and dynamic properties of the object, caused by the effect of biological materials responding differently to an excitation signal of varying frequency [1]. The method of EBI is highly suitable for integration into wearable devices due to the simplistic circuitry and low power consumption. Providing several features, including the feature extraction of hemodynamic parameters, the full potential of EBI has not yet been fully utilized in biomedical signal acquisition [1]. The reason is presumably lying in the proneness of the method to the movement of the sensor relative to the body surface, i.e., the uncertainty of the measurements.

EBI is not the only suitable method for the wearable monitoring of peripheral arteries and cardiovascular parameters: other techniques also give promising results, such as PPG [4], magnetic induction [5], accelerometer-data-based ballistocardiography [6], piezoresistive pressure sensors [7], etc. However, the property of EBI to issue the changes in blood volume that, among the shape of the waveform itself, reflect the cardiac cycles with vital hemodynamic fiducial points, is an advantage. The waveform of EBI, acquired from the radial artery, has been shown to reflect the changes in the central aortic pressure (CAP) of blood [8]. Hence, the EBI waveform and the respective impedance cardiogram are a source for noninvasive blood pressure detection solutions [4,9,10].

The monitoring of the signal of EBI [11] and PPG (an excellent review is available in [12]) from peripheral locations has been reported in the literature before. While the non-invasive measurement of EBI is focused generally on a certain artery [13,14] or the whole body [15], the PPG can be acquired from anywhere where the perfusion of the circulation is present. In the current work, the the signal of reflection mode PPG (rPPG) signals were considered as references to the EBI. Therewith, joint acquisition applications from certain and targeted arteries are rare, constituting the novelty of the current research. It is worth noting that the current paper is not dealing with disease detection or diagnosing—the content is purely technical.

The pulse wave is an important variable based on which a variety of health-related indicators can be derived. The amplitude, area, and time features of the signal have been utilized, among others, for estimating the blood pressure based on the EBI [2,4], chronic kidney disease based on the ECG [16], and arterial stiffness based on the PPG [12]. Therefore the quality of the signal is of high importance—defined in the current paper as the continuity. In real-time applications, the selection of sequences of the acquired, randomly partially interfered signals is often a problem [17]. How to classify and sort these usable signal sequences in a case where the signal morphology is a matter of change–especially when the acquisition site may differ? At which level does the signal morphology vary from artery to artery and what are the most suitable locations on the body surface for joint acquisition of the signals of EBI and PPG? For this, a deep-learning-based model, imposed specifically on a small amount of data, was applied.

A deep-learning-based model was proposed and applied in the current work to evaluate the gathered signals of EBI and rPPG. However, it was considered as a tool used to assess the data, determine the significant classifiers, and provide data for comparisons and conclusions. Deep learning models have been proposed in the literature to determine the activity of a man based on inertial sensors [18], vascular aging prediction based on a PPG signal [19], etc. Ref. [20] constitutes the closest solution to our proposed model, classifying a pulse wave by five CVDs based on the blood pressure curve. The similarity is evident—five certain pathological pulse wave patterns—as specific waveforms of EBI signals acquired from different arteries. However, the model in [20] is applied to a relatively large amount of data (412 subjects), while, in our model, a small-data-based approach (data of a single subject) is used. Moreover, in [20], five predefined pulse waveforms corresponding to certain pathological conditions were used to train the convolutional neural networks (CNN) model, while, in our case, the data were derived from the available small data.

This paper introduces the joint acquisition options of pulse waves from five selected arteries by using the techniques of EBI and PPG. Moreover, four different excitation frequencies were considered in concurrent EBI signal acquisition. Together with EBI, rPPG was picked up, providing means for direct comparison and consideration to be applied in sensor-fusion-based applications. A qualitative and quantitative evaluation of the signals was performed to assess the suitability of the selected locations for the wearable monitoring of peripheral artery EBI and PPG. The most suitable locations for EBI and rPPG-based pulse wave monitoring are proposed, from which the posterior tibial artery constitutes a novel solution.

The goal of the performed research was to determine the suitable arteries (and respective skin sites) for joint EBI and PPG signal acquisition toward wearable regional impedance cardio- and photoplethysmogram generation. While the latest publications focus on the arm [21,22], there are peripheral arteries whose potential is experimentally unexplored and compared. Our choice of arteries was based on practical considerations, i.e., the potential of the skin site for wearable device connection (e.g., in the form of a wristwatch, necklace, sock, sleeve of a shirt). The research aimed to provide options for attaching the signal acquisition module(s) (electrodes and sensor unit) on the body surface for projecting and designing wearable devices that possibly issue signals of improved quality for processing and decision making. The availability of cardiac-related information in pulse wave signals relies on the quality of the measured data, maintaining their morphological properties—a base for the medical decision-making process. This study does not propose projecting and design solutions for wearable devices, but focuses exclusively on the acquisition options of the signals. The motivation for the research was to explore possibilities for the unobtrusive joint monitoring of the EBI and PPG of peripheral arteries for next-generation pulse wave signal morphology-based wearable devices.

## 2. Background and State of the Art

In this section, the background art of the research topic is presented. The regional impedance measurements are described, followed by an introduction to small data deep learning.

### 2.1. Features of Regional Impedance Measurements

Depending on the location of the peripheral artery in the arterial tree, the shape of the pulse wave varies. Such phenomena are related to the properties of the cardiovascular system, where the central arteries are more elastic than the peripheral arteries [23]. Due to that, the systolic peak gains a more narrow geometry, and its amplitudes are increased. As a practical outcome, systolic blood pressure can be of higher value in peripheral locations.

The described phenomenon also applies to the waveform of impedance, which gains varying shapes depending on the acquisition site, i.e., the targeted artery. The impedance measurement approach for targeting the peripheral arteries in hemodynamic fiducial point detection is called regional impedance cardiography (RICG) [14]. The RICG method may be considered the most convenient solution for wearable monitoring instrumentation, enabling miniature device design and a dense arrangement of electrodes. The wrist has been shown to possess the highest potential as a place for wearable devices in several publications through being the traditional location of a wristwatch. Thereat, the electrodes may be targeted to the ulnar [24] and radial artery [9,25] or both [10,26]. The relative closeness of the carotid artery to the aorta has shown good results in pulse sensing through EBI measurement [27]. Also, the brachial artery has been considered, e.g., in EBI-based cardiac output and stroke volume estimation [28]. Other peripheral arteries are rarely defined as targets for EBI measurement applications.

A large and controversial issue is the electrode positioning on the skin surface relative to the changing volume in an artery underneath the skin. The exact sensitivity distribution in the biological matter is still debatable [29] and largely unpredictable. It has been shown in the literature that specific (like focused impedance method) and shifted electrode configurations relative to the artery could perform better [30]. However, the current research relies on a standard in-line four electrode configuration to be comparable with the vast amount of published research reports (also of our research group [13,31]).

### 2.2. Small Data Deep Learning for Biomedical Signals

The classical prerequisite for the deep learning model application is the presence of a large amount of data—the larger, the better. However, this is not always possible in real-time monitoring—especially when considering a first-time application of a wearable device on a subject. Among others, there are solutions for small data scenarios, like transfer learning, data augmentation (i.e., synthetic data generation), and autoencoders.

Transfer learning has been successfully reported in the literature for the automatic interpretation of ECG [32,33]. However, in the case of transfer learning, a large amount of data is still needed for training the model. If the model is trained on sufficiently rigorous data (including all the possibly appearing features), it can be used for small data (e.g., in real-time application).

Data augmentation is proposed as a possible solution for small data, especially when there is a lack of annotated data [34]). Used originally in image classification (an excellent review of methods is available in [35]), the approach can also be extended to biomedical signals like ECG [36,37,38] and PPG [39]. The most known data augmentation method for training the CNN is generative adversarial networks [40]). In essence, during data augmentation, the mass of data is increased and diversified through synthesizing. The synthesized data are based on the existing ones but are mixed with random artificial perturbations. However, the augmentation method is limited by the available amount of data.

Yet another way is the application of autoencoders, from which variational autoencoders (VAEs) have gained attention in the literature lately for biomedical signals [41]. In summary, VAE is an architecture that relies on a probabilistic description for observing the latent space formed through the encoding and decoding of the data [42]. The goal of VAE is to generate similar data from random features that are gained from the small data by compressing the high-dimension input to its lower-dimension representations [10].

However, none of the described methods propose a 100% working solution for small data deep learning. Pre-knowledge of the data or an existing database containing all possible samples is still needed for the best outcome. However, for small data, tailor-made solutions based on the above-described deep learning methods are highly suitable.

## 3. Materials and Methods

In this section, firstly the measurement devices used in the performed experiments are listed and characterized. Secondly, the applied methods for signal acquisition and processing are presented, together with the description of the proposed CNN architecture.

### 3.1. Measurement Devices

The measurements were performed by an impedance spectroscope HF2IS with the trans-impedance amplifier HF2TA of Zurich Instruments AG (Zurich, Switzerland). This device enables simultaneous impedance measurement at four frequencies with a maximum of 50 MHz and is supplied with two analog auxiliary and two digital inputs [43]. The device is suitable for sensor-fusion-based biomedical signal acquisition.

Monitoring electrodes 2228 of 3M (Maplewood, MN, USA) were utilized for picking up the impedance signal. For monitoring the rPPG, we used the PulseSensor of World Famous Electronics LLC (New York, NY, USA). The device is formulated on a printed circuit board (PCB) and relies on a green-light-emitting diode (LED). While the penetration depth of green light into the tissue is shorter due to the strong absorption by hemoglobin, it is reflecting a clearer indication of blood volume [44]. Green LEDs are suitable for acquiring the PPG signal in locations where blood perfusion reaches the skin surface.

### 3.2. Measurement Method

Five arteries in the human body were selected for monitoring the signals of EBI and rPPG based on the literature and the author’s practical experience. The main requirements for the selection were:Closeness of the artery to the skin surface—enabling compact electrode configurations;Possibility of attaching a wearable device (sensor(s)) to the selected acquisition site.

The following arteries (and respective body areas) were selected for evaluation (Figure 1): brachial artery (arm) (A), carotid artery (neck) (B), radial artery (wrist) (C), posterior tibial artery (leg) (D), and ulnar artery (wrist) (E).

The original size of the electrodes was modified to a width of 10 mm. For measuring the impedance, the electrodes were attached to the skin surface, approximately on top of the location of the selected artery. We used only an in-line four-electrode placement strategy with a distance of 5 mm between the electrodes (where the outermost pair was for excitation and the inner pair was for measurement). The rPPG sensor was always attached close to the outer (exciting) electrode at an approximate distance of 5 mm.

For measuring the EBI (*Z*), an excitation signal with an amplitude of 500 mV and frequencies of 50, 100, 500, and 1000 kHz was applied to the object. From the chosen values, the first two fall into the standard range for non-invasive measurement of EBI [45]. We included the higher frequencies for comparability reasons as the effect of different excitation frequencies on the morphology of the EBI signal can be assumed [1].

For the characterization of the acquired signals, the approach of calculating the time-varying change of impedance (Δ*Z*(*t*)) was applied—represented by Equation (Equation 1). Specifically, the *Z* of a body can be expected to consist of the invariable impedance of the local volume $Z_{0}$ and Δ*Z*(*t*), caused by pulse wave [13] (expressing in Ohms (Ω)): (1)$$Z=Z_{0}+\Delta Z\left(t\right).$$

The time-varying change in Δ*rPPG*(*t*) was defined and calculated by using Equation (Equation 2) similarly: (2)$$rPPG=rPPG_{0}+\Delta rPPG\left(t\right),$$
where $rPPG_{0}$ can be considered to be the invariable state of the scattering of light in the body tissue (expressed in volts (V) as read from the sensor output).

The experimental procedures involving humans, described in the current paper, strictly follow the principles outlined in the Helsinki Declaration of 1975 as revised in 2000.

### 3.3. Deep-Learning-Based Method and the Proposed CNN-Architecture-Based Model for Pulse Wave Signal Evaluation

In the current work, a deep-learning-based model for the evaluation of the gathered signals of EBI and rPPG was composed. As the amount of gathered data was relatively small, and no annotated sets of signals of ICG and rPPG were available, the transfer learning approach was not used. Instead, we relied exclusively on the present data logs of relatively small amounts by applying a suitable deep learning approach.

An approach for increasing the amount of data, especially in the case of periodic waveforms (like cardiac signals), is to consider the signals as a series of periods instead of the full acquired signal length (based on some feature—like a distinctive peak (e.g., R wave in ECG waveform)). In this work, the maximum systolic flow, i.e., the C point of the signal of ICG and the systolic peak of the signal of PPG, was considered as such features.

Also, the cardiac cycle length varies because of physiological, emotional, physical, etc., reasons, and the period lengths of the signal waveform are varying. As a solution, we propose the application of a window of specified length (calculated based on each waveform in each case) in which the midpoint is defined by the moment of maximum systolic flow. In practice, the window is shifted either forward or backward, depending on the locations of maximum systolic blood flow peaks—so overlapping appears.

As follows, these individually defined periods, based on the proposed window function, were considered separate pieces of data. Additionally, all periods were labeled with the information of the acquisition site (artery).

In the case of the proposed deep learning model, the signals were conditioned during the processing. Specifically, a Butterworth band-pass filter of order 2 with cut-off frequencies of 0.5 and 50 Hz, respectively, was applied. For processing, all the signals were normalized through the following two-step approach:The mean value was removed, i.e., set to zero by subtracting from all of the values;The unit variance was found by dividing the values by the standard deviation (SD), marked as σ.

The model was composed in Python 3.11 by using *sklearn*, and 5-fold cross-validation was applied to set the weights of training/testing to the ratio of 4:1. The general workflow of the proposed method is visible in Figure 2.

The proposed deep-learning-based model was then applied to all of the defined periods of the signal. For each case, a confusion matrix was generated to estimate the accuracy and probability.

Since we have a classification problem (not binary classification) and the data were fully balanced, we used a simple approach—illustrated by Equation (Equation 3). The accuracy (*Acc*) was calculated according to (denoting the percentage of classifying the signal correctly):(3)$$Acc=Pred_{correct}/Pred_{total}$$
where $Pred_{correct}$ denotes the number of correct predictions and $Pred_{total}$ denotes the total number of predictions, i.e., we classify all periods and finally choose the one that is predicted the most (maximum function)—as a result, we classify the original signal.

Also, the probability was calculated in each class (data set acquired from each chosen body site (artery)) by showing the proportion of correct predictions from the total number of predictions. The probability, in the context of this paper, is the percentage of instances in which we have classified the periods correctly. It is calculated similarly to accuracy, but instead of finding the ratio of final predictions of the signal, we compared the ratio of period predictions.

#### Proposed CNN Architecture for ICG and PPG Signal Evaluation

CNN is a deep-learning-based neural network architecture that contains multiple layers—inspired by the structure of the human brain [46]. The basic idea of CNN is that the data in a model are transferred from layer to layer, whereas the trivial features are extracted in the first layers and complex features in the deeper layers.

In this study, the model was developed using the Python programming language and the TensorFlow framework with Keras as its high-level application programming interface (API), running on the cloud-based development environment provided by Google Colab. The underlying operating system used was determined to be Linux. The simulations were run on a standard machine without using graphics processing units (GPUs) or tensor processing units (TPUs). The architecture with the parameters of the proposed CNN is visible in Table 1 and infrastructure in Figure 3.

The proposed model has nine layers (pooling layers can be omitted), of which six have trainable parameters. The layers can shortly be described as follows:Starting with average pooling (averagePooling1D)—as the initial data sample rate is high (1666 Hz), which is not necessary in EBI and PPG signal feature extraction, it is downsampled by averaging five values;One-dimensional convolution layer (Conv1D) with a kernel size of 4 for picking up the trivial features—activated by the rectified linear activation function (ReLU);Pooling layer (MaxPooling1D) for reducing the dimensions of features map;Convolution layer (Conv1D) with a kernel size of 20 for extracting more abstracted features;Pooling layer (MaxPooling1D) for further reduction in dimensions;Flattening of the previous layer (Flatten)—although the input is one-dimensional, an empty layer has been added previously to continue with fully connected layers;Fully connecting layer (Dense);Reduction in the overfitting by using dropout layer (rate of 0.3) (Dropout);Fully connected output layer (Dense) with a length dimension of 5 as the data have five different classes.

Total and trainable parameters by the model: 4419 (non-trainable parameters: 0).

## 4. Results

Three consecutive measurements were performed at each chosen sensor location, resulting in three data logs of EBI and rPPG in the time scale. The analog signal was sampled by quite a high frequency—1666 Hz—to possess the opportunity of restoring the signal as rigorously as possible. While the rPPG signal was relatively clean and free from interference, the EBI signal required filtering. For the detection of Δ*Z*(*t*) and Δ*rPPG*(*t*), the signals were post-processed in LabView software by using a Savitzky–Golay filter at side points of 100.

The sample signal waveforms of EBI and rPPG, acquired from all five chosen locations on the skin surface at an excitation frequency of 100 kHz are visible in Figure 4a–e.

For quantitative evaluation, the Δ*Z*(*t*) and Δ*rPPG*(*t*) were calculated for all three acquired signals of EBI and rPPG from each chosen position, constituting an essential quality measure. In Table 2, the mean values (Δ*Z*(*t*)*_mean_* and Δ*rPPG*(*t*)*_mean_*) together with SDs (σ) are shown.

An even more informative variable is the sensitivity (Equation (Equation 4))—the ratio of Δ*Z*(*t*) from the maximum peak value of *Z*. Sensitivity can be expressed as
(4)Sensitivity=(ΔZ(t)×100)/Z,
showing the presence of Δ*Z*(*t*) caused by pulsating blood in the artery in correlation to the total measured value of *Z* at a chosen electrode location. The calculated sensitivities of the measured signals of EBI and rPPG in the cases of all the chosen signal acquisition sites can be seen in Figure 5a,b, respectively.

As the confusion matrix result is different in each application of the model (the random choice of signal periods for training and testing differs), the median of 10 runs was considered for the EBI and rPPG signals separately. In the case of EBI, separate runs were performed in the cases of all four applied excitation signals to verify the effect on the morphology of the acquired signal. Based on the gained confusion matrices, the average accuracy and probability were calculated in the cases of each excitation frequency. The results are visible in Table 3.

A representation of the EBI and rPPG signal periods, selected by the deep-learning-based model in the cases of five selected acquisition sites (arteries), is visible in Figure 6a–e.

The average probability, calculated by the proposed CNN architecture in each of the five cases (data logs), is available in Table 4.

## 5. Discussion

As expected, the comparison between the two chosen sensor types for peripheral artery monitoring resulted in rather distinctive results. The principle of the chosen sensors differs, i.e., the EBI indicates the opposition of the matter to the applied electrical stimuli while the PPG denotes the amount of light either transmitted or reflected from the skin. As a result, their performance varies in different skin sites, affected by anatomical properties (e.g., fat layer thickness, the closeness of large muscle volume, etc.).

While it is well known that the Δ*Z*(*t*) is fairly low (but consistent and measurable) in the case of the radial artery (e.g., [13]), the performance of the posterior tibial artery was a surprise. This is contrary to [11], where the performance of radial and ulnar arteries outperforms the tibial and carotid arteries. The reason lies in electrode dimensions and exact targeting to the source of the signal (pulsating volume of blood), but most importantly the choice of the tibial artery (posterior tibial in the current research in contrast to anterior tibial in [11]).

The highest measured Δ*Z*(*t*) (Table 1) and sensitivity (Figure 5a) in the case of the carotid artery are explainable by the presence of several arteries in the neck: internal and common carotid arteries (CCAs) and the vertebral artery. The diameter of, e.g., the CCA (mean value of 6.52 ± 0.98 mm [47]) is two times larger than that of, e.g., the radial artery (mean value of 2.62 ± 0.24 mm [48]). We explain the poor performance of the brachial artery by the anatomy. The brachial artery is located close to the biceps muscle, whose conductivity is only twice lower (0.362 S/m at 100 kHz [49]) than that of arterial blood (0.703 S/m at 100 kHz [49]). In the cases of radial, ulnar, and posterior tibial arteries, the bone is located near the measurement site while, in the case of the brachial artery, the large muscle volume is expected to shunt the measured modulations carried by the arterial blood.

The measured value of Δ*Z*(*t*) in the case of the chosen peripheral arteries is generally in the order of magnitude with the published data. Though being highly dependent on the exact placement of the electrodes, the comparison can be performed. For the radial artery, Ref. [24] has declared the Δ*Z*(*t*) in the range of 0.045–0.119 Ω and, for the ulnar artery, in the range of 0.048–0.131 Ω. Classically, the electrode placements for brachial artery EBI monitoring are not densely located, and the Δ*Z*(*t*) varies on a larger scale in the published works [50]. The same applies to the carotid artery, where our measured Δ*Z*(*t*) exceeds the values derived, e.g., in [27]. For the posterior tibial artery, respective comparative measurement results of EBI are unavailable, constituting an unexplored area.

Also, the effect of respiratory activity is visible in the cases of certain acquisition sites (brachial and carotid arteries (Figure 5a,b)). In the case of the carotid artery, the air directly passes through the adjacent pathway in the throat. In the case of the brachial artery, the effect is more related to physiology. The human body is one entity, and all the processes are relevant—in- and exhalation influence the circulatory system. Specifically, during in- and exhalation, the pulmonary arterial end-diastolic pressure is changing accordingly, affecting the pressure in the pulmonary artery [51]. The effect is revealed as modulations on pulse waves—carried throughout the circulatory system. Respiratory activity can also be measured from the wrist by using a wearable device [52].

The essential observation in comparing the performance of both used sensor types is the higher sensitivity of the signals of rPPG compared to the EBI (Table 1). However, through the prism of sensor fusion, both signals are required to determine, e.g., the PTT, and estimate the blood pressure [9]. While in the case of the EBI signal, the measurement result of Δ*Z*(*t*) is very electrode-placement-specific, it is not so critical in the case of rPPG. For detecting rPPG, the presence of the perfusion of blood flow close to the skin surface is required—and unlike the signal of EBI, the perfusion in the tissue is enough [44].

In the case of applying the developed deep-learning-based model for biomedical signal evaluation, two questions appear: (1) how does the signal morphology (e.g., the depth of modulation in the carrier signal) differ in different arterial locations and (2) what is the effect of processing (filtering, conditioning, etc.)? Based on the practical measurements of our research group [22] and the literature [11], the morphology of the EBI signal changes depending on the acquisition site. The change can be visually assigned [22] and is expected to correlate with the pressure wave [31].

The signal of rPPG is more consistent through all of the chosen locations; however, its morphology is expected to be acquisition-site-dependent. The processing of the signal affects its morphology, e.g., dignifying some characteristic points (like differentiation) or removing distinctive frequencies (like filtering). Thus, the deep-learning-based models perform best if the signals under consideration are processed equivalently.

The excitation signal frequency affects the measured signal of EBI (Figure 6), or, more precisely, its origin—the contribution of biological matter that the signal represents. The average accuracy, found by the proposed deep-learning-based model, shows a rising trend—with the increase in excitation frequency, the average accuracy increases (Table 3). However, in the case of the sensor location-specific measure—average probability—the outcome is not so distinct. Nevertheless, the result seems to support the suggestion to choose an excitation signal for EBI measurements close to the frequency of 100 kHz [45] (refer to Figure 6b and Table 3). Meanwhile, the signal of rPPG shows the highest value, exhibiting a more stable and extinct pattern and providing the best signal quality (Figure 6e).

A term continuity of the signal may be presented based on the results of the applied deep-learning-based model, denoting the signal property used to maintain its morphological features throughout consecutive electrical measurements. The modulation depth on the signal plays an important role, illustrated by the EBI waveform that is acquired from the neck (carotid artery) (Figure 4b). The calculated mean values of the average probability indicate the highest level of continuity for the carotid artery in front of the ulnar artery, tibial artery, and radial artery (Table 4). The brachial artery competitively shows the lowest continuity (Table 4). Based on the continuity criteria in the case of EBI exclusively, the carotid artery could be the best location for cardiovascular signal acquisition.

In relying on the multisensory domain, i.e., adding the functionality of PPG, the posterior tibial artery exhibits considerable results. The posterior tibial artery is located close to the skin surface, next to the calf and shinbone—providing prerequisites for the emergence of the sensitivity field on the pulsating volume of blood in the artery. The tibial artery has been considered as a possible target in pulse wave detection before [11], but has not been classified among the best ones.

While the proposed deep-learning-based model needs testing on a large sample size, its performance on small data is promising. The EBI-based pulse waveforms differ slightly, depending on the acquisition distance from the heart (but also on the exact artery). The gained average probability values (Table 3) are located close to the 50% level, depending on the excitation frequency. Moreover, as the signals are not pre-processed (except for band-pass filtering), the results are noticeable (Table 4).

The average accuracy gained by our model is comparable to the closest solution from the literature [20], being one percent higher (0.95 vs. 0.96) at a 1000 kHz EBI excitation frequency. However, the EBI excitation frequency effect and the higher determined continuity of the rPPG signal resulted in varying results. Most importantly, in this research, a small data approach was used, compared to relatively large-scale data in [20].

The solution proposed in the manuscript differs from the existing literature in terms of a joint acquisition of EBI and PPG by using a location-sharing approach. PPG is classically included as a reference from the index finger (e.g., in [11]) and not from the shared skin site with EBI electrodes. In this study, the results have been presented through the perspective of both variables, indicating the most suitable skin sites for a location of possible sensor-fusion-based wearable devices. Even more, based on our best knowledge, there are no published studies where the posterior tibial artery would have been exclusively targeted and experimentally proved as a possibly promising location for joint EBI- and PPG-based peripheral artery monitoring. The proposed solutions have included either the anterior tibial artery [11], dorsalis pedis artery in the foot [53], or the whole foot plantar arteries [54].

## 6. Limitations of the Work

The presented research is limited in certain aspects that need attention in considering the gained results. The signals were acquired only by involving a single volunteer. As a result, the lack of inter-individual variability appears, requiring a larger sample size for possible extended research. However, performed as a pilot study, the results are acceptable and considerable for publishing.

The presented and applied deep-learning-based model needs verification on a larger amount of data to verify its accuracy and probability. Moreover, the model needs testing on different formats of biomedical signals; for example, in the context of EBI (dZ/dt), its first derivative—commonly known as the ICG. As in the case of dZ/dt, the peaks in the signal of ICG become narrower—the same appears when differentiating the signal of rPPG—and the signal morphology will change, possibly affecting the performance of the deep learning model.

## 7. Conclusions and Future Work

We have verified the presence of pulse waves in acquired signals of EBI and rPPG from all five chosen body sites (peripheral arteries). The most significant outcomes of the research are listed subsequently:For Δ*Z*(*t*), the carotid artery exhibits the highest sensitivity, while the posterior tibial artery outperforms all other locations in rPPG detection. In the case of summing the sensitivities of EBI and rPPG, the suggested location for a joint EBI and PPG acquisition-based wearable device would be the posterior tibial artery;The best average performance, i.e., continuity, throughout the four chosen excitation frequencies in the case of a joint acquisition of EBI and rPPG was experimentally proved to be delivered by the posterior tibial artery;The best average probability is provided when utilizing a classical excitation signal frequency of 100 kHz.

The average accuracy presents a rising trend—improving continuously with an increasing excitation frequency. However, the performance of PPG exceeds the similar property of EBI—providing an average probability of 77.22%. The best average performance, i.e., continuity, throughout the four chosen excitation frequencies is delivered by the carotid artery. Also, the help of the proposed model in signal processing has been significant, revealing the increasing continuity with an increasing EBI excitation signal.

This research gives significant insight into the electrode location selection, which classically has been inclined to the wrist and especially to the radial artery. When considering the results of the presented research, the posterior tibial artery forms a strong candidate for the acquisition site of EBI and rPPG signals in a wearable device.

The presented research can be extended in certain aspects, especially toward the multisensory acquisition of cardio-respiratory data. In addition to the EBI and PPG, the ECG signal acquisition could provide added value (e.g., the determination of a pre-ejection period)—also shown by our research group to be available from non-standard body areas [55]. Moreover, by further developing the proposed deep learning model, signal-waveform-based disease prediction or person recognition could be a target.

The proposed deep-learning-based model for signal evaluation needs verification on a large amount of data. The target is to extend the experimentation by developing a comprehensive testing scheme and recruiting more volunteers. Another option that we consider is to test the proposed model on data in public repositories.

## Figures and Tables

**Figure 1 sensors-23-07111-f001:**
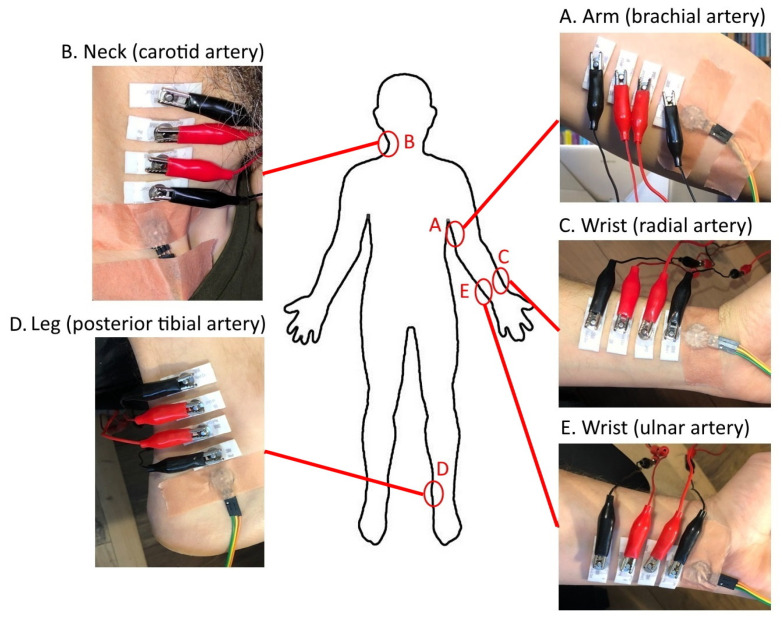
Locations of the chosen arteries and respective body areas for acquiring the signals of EBI and rPPG: brachial artery (arm) (**A**), carotid artery (neck) (**B**), radial artery (wrist) (**C**), posterior tibial artery (leg) (**D**), and ulnar artery (wrist) (**E**).

**Figure 2 sensors-23-07111-f002:**
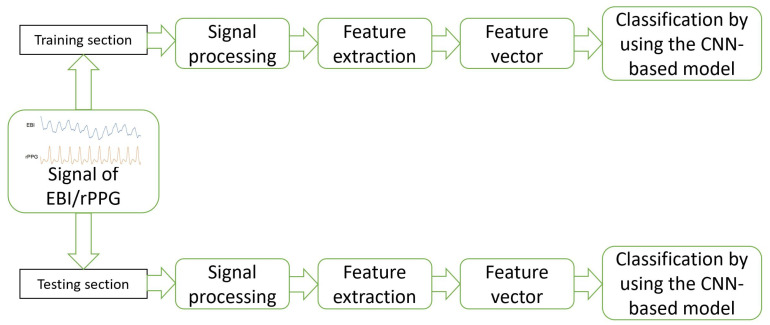
Workflow of the proposed method and deep-learning-based approach for the EBI and rPPG signal evaluation.

**Figure 3 sensors-23-07111-f003:**
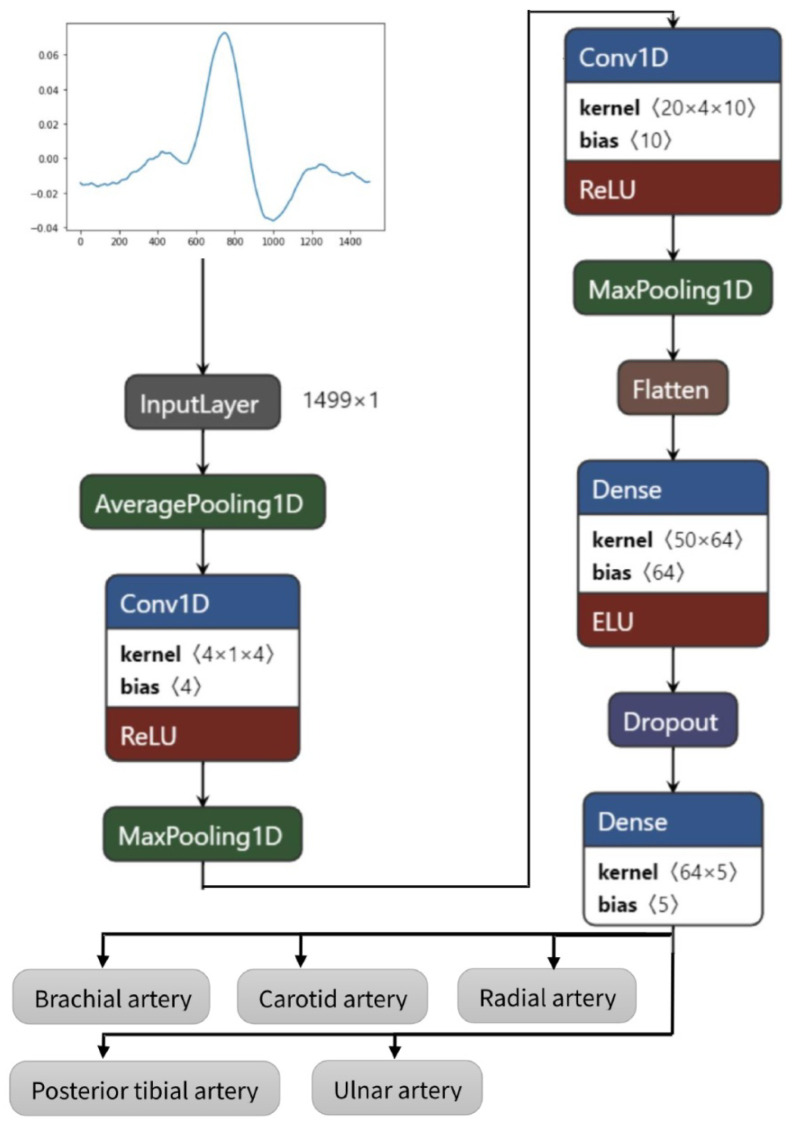
Convolutional neural networks architecture of the proposed model.

**Figure 4 sensors-23-07111-f004:**
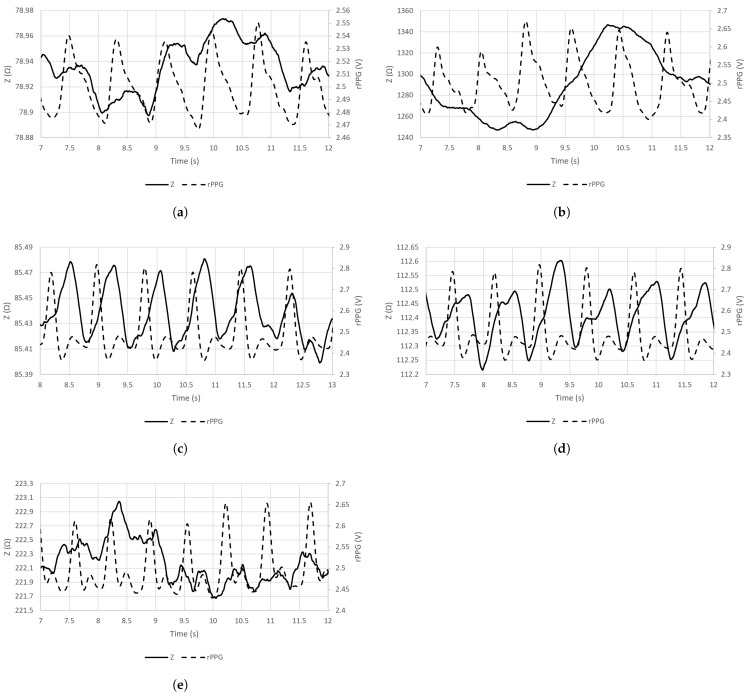
Sample waveforms of signals of EBI (at the excitation frequency of 100 kHz) and rPPG that were acquired from the chosen sensor locations (A–E). (**a**) Brachial artery (A); (**b**) carotid artery (B); (**c**) radial artery (C); (**d**) posterior tibial artery (D); (**e**) ulnar artery (E).

**Figure 5 sensors-23-07111-f005:**
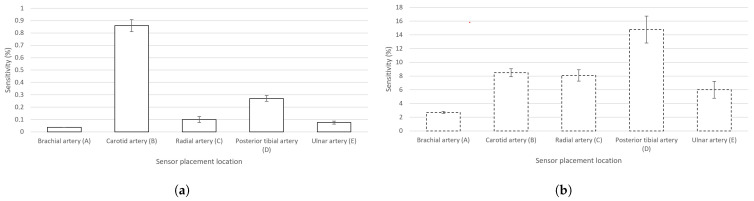
Calculated sensitivities (mean values) (with graphically added σ) of the EBI and rPPG signals in the cases of the chosen sensor locations (A–E). (**a**) Signal of EBI; (**b**) signal of rPPG.

**Figure 6 sensors-23-07111-f006:**
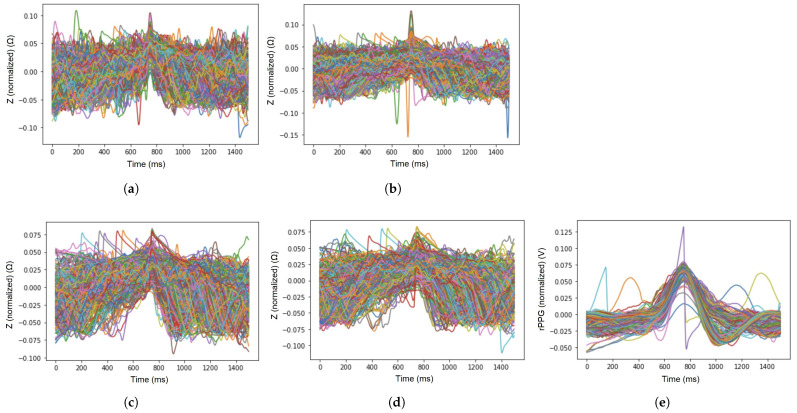
Signal periods (marked with different colors for each period—but not defined as the figure is illustrative) that were represented by the proposed deep learning-based model for evaluation of the signals of EBI at each excitation frequency. (**a**) Signal periods of EBI at 50 kHz, (**b**) signal periods of EBI at 100 kHz, (**c**) signal periods of EBI at 500 kHz, (**d**) signal periods of EBI at 1000 kHz, and (**e**) signal periods of rPPG.

**Table 1 sensors-23-07111-t001:** Proposed CNN architecture for the EBI and rPPG signal evaluation.

Layer Type	No. Channels	Length Dimension	No. of Parameters
AveragePooling1D	1	299	-
Conv1D	4	296	20
MaxPooling1D	4	74	-
Conv1D	10	55	810
MaxPooling1D	10	5	-
Flatten	-	50	-
Dense	-	64	3264
Dropout	-	64	-
Dense	-	5	325

**Table 2 sensors-23-07111-t002:** Results of calculated Δ*Z*(*t*) and Δ*rPPG*(*t*) with σ in the cases of chosen locations of the sensors.

Signal Acquisition Site	Δ*Z*(*t*)_*mean*_ ± σ (Ω)	Δ*rPPG*(*t*)_*mean*_ ± σ (V)
Brachial artery (A)	0.029 ± 0.0005	0.068 ± 0.0039
Carotid artery (B)	11.123 ± 0.1500	0.225 ± 0.0164
Radial artery (C)	0.085 ± 0.0229	0.216 ± 0.0227
Posterior tibial artery (D)	0.304 ± 0.0253	0.413 ± 0.0598
Ulnar artery (E)	0.167 ± 0.0279	0.158 ± 0.0336

**Table 3 sensors-23-07111-t003:** Average accuracy (Av. Acc.) and probability (Av. Prob.) in the cases of the acquired signals of EBI (at the four chosen excitation frequencies) and rPPG.

Signal	Av. Acc. (%)	Av. Prob. (%)
EBI at 50 kHZ	70.00	45.91
EBI at 100 kHZ	80.00	52.35
EBI at 500 kHZ	84.00	43.35
EBI at 1000 kHZ	96.00	47.02
rPPG	98.00	77.22

**Table 4 sensors-23-07111-t004:** Average probability (Av. Prob.) in the cases of the EBI and rPPG signals acquired from the five selected acquisition sites (arteries) at the four chosen excitation frequencies.

Signal Acquisition Site	Av. Prob. of EBI at 50 kHZ (%)	Av. Prob. of EBI at 100 kHZ (%)	Av. Prob. of EBI at 500 kHZ (%)	Av. Prob. of EBI at 1000 kHZ (%)	Av. Prob. of rPPG (%)
Brachial artery (A)	23.15	16.67	20.74	32.50	77.41
Carotid artery (B)	50.19	64.81	59.07	63.52	61.20
Radial artery (C)	57.13	45.65	48.52	41.39	71.48
Posterior tibial artery (D)	36.57	54.72	53.06	53.70	97.87
Ulnar artery (E)	62.50	79.81	35.37	43.98	78.15

## Data Availability

The data presented in this study are available on request from the corresponding author. The data are not publicly available due to the privacy restrictions.

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
