# Peer review of "Sensor-Location-Specific Joint Acquisition of Peripheral Artery Bioimpedance and Photoplethysmogram for Wearable Applications"

_sensors, 2023, doi:10.3390/s23167111_

Round 1
Reviewer 1 Report (New Reviewer)

Author Response
Please see the attachment.

Reviewer 2 Report (New Reviewer)
This paper describes the finding of suitable measurement sites of EBI and PPG. It is difficult to follow this study, because we could not find a final goal and outcome of this study. If this research designed the development of a combined EBI and PPG monitor, the detecting parameters must be considered. In this study, only blood volume changes are detected. EBI can detect respiratory parameters in thoracic impedance measurement and changes in blood flow in peripheral measurement. While PPG waves show the blood volume changes. Currently, both methods were used as the estimation of blood pressure.
The experiment is clear but unfortunately, we cannot find any novelty of these results.
From these results, the authors propose a new device with combined EBI and PPG.
Minar comment;
EBI is based on usually cylinder model. The dimension of measurement sits strongly affected the electric current distribution.
Author Response
Please see the attachment.

Reviewer 3 Report (New Reviewer)
The authors present their experiments conducted by location-specific sensors of electrical bioimpedance (EBI) and photoplethysography (PPG). Five locations of the human body are monitored related to five arteries respectively. The acquired signals are evaluated in terms of sensitivity gain and continuity. Denoting by continuity the signal property to maintain its morphological features it is posed as a classification problem which measures the distinctiveness of location-specific signals.
The following issues should be addressed prior to publication:
Major
• The results are not related to the cardiac activity or the early detection of CVD. The main task seems to be the investigation of the distinctiveness of location-specific signals without any straightforward diagnostic conclusion related to the cardiac activity. For clarity reasons, it should be stated from the beginning that the experimental results do not support any discussion with regard to any diagnosis or early detection and the words “monitoring of Cardiac Activity” should be removed from the title.
• In Figure 2, the training process and the testing process are presented as two converging paths of feature extraction. Those are strictly distinguished procedures. The output is a classification result as well and should be noted as such.
• In section 3.3 is mentioned that the probability is calculated in each class. Please clarify the meaning of probability. Do you have a softmax output in your CNN classifier? This seems to be critical for the presentation of the results.
Minor
• In Fig.5 the calculated sensitivities into parentheses are noted as σ but in equation 4 as S.
Round 2
Reviewer 2 Report (New Reviewer)
This paper describes the basic study of Bioimpedance and PPG. The reviewer does not clear the aim of study. however, the experiments and results are clear. The reviewer wants to know the final proposed wearable device.
Author Response
Please see the attachment.

Reviewer 3 Report (New Reviewer)
The authors addressed my comments accordingly making more clear the task of their work and the interpretation of the results.
The manuscript is recommended for publication.
Author Response
Please see the attachment.

This manuscript is a resubmission of an earlier submission. The following is a list of the peer review reports and author responses from that submission.
Round 1
Reviewer 1 Report
The authors have attempted to explore the use of ECG and PPG measurements from multiple peripheral anatomical positions and implemented a CNN model for site acquisition using a small amount of data. My major comments are as follows:
The motivation for using ECG and PPG measurements for sensor site acquisition is not clearly explained. The authors should provide a clear reason for why this information is not known or prescribed, and what exact new information is extracted and requires a complex modeling. For example, if the model helps to precisely identify the distance between the on-skin sensing sites and the underlying artery, this would be a valuable contribution. The 5-way classification presented for the visually identifiable sites appears to be a contrived problem. The authors should consider investigating if there is any additional, latent information encoded in the CNN model that could be used to predict unseen outcomes, which would be an interesting area for future research.
The manuscript also lacks a clear testing scheme for evaluating the utility of the developed CNN model across multiple subjects and anatomical locations. A more comprehensive investigation should be implemented to validate the results and demonstrate the performance of the developed CNN model.
In conclusion, the authors have reported ECG and PPG measurements from multiple peripheral anatomical positions and implemented a CNN model for site acquisition. However, the motivation for the study and the validity of the results are not well-supported. A comprehensive testing scheme across multiple subjects and anatomical locations is needed to validate the effectiveness of the model.
Reviewer 2 Report
- The current article refers to a system for monitoring cardiac activity using biomedical signals, specifically designed for wearable applications, and able to adjust to the location of the user. This technology would allow for continuous monitoring of a person's heart activity, potentially providing early detection of cardiac issues, and allowing for more accurate tracking of cardiac health. The "smart" aspect of the system would refer to its ability to adjust and adapt to the user's location, providing more accurate and relevant information.
- The abstract needs to be slightly improved. From the abstract, it is better to summarize the limitations of existing works to highlight the necessity of the study given that this topic is not new. From the abstract, I cannot find any novel points about the solution. There is no description of how the verify the effectiveness of the proposed solution. Finally, no concrete experiment results demonstrate the advantages of the proposed solution.
- The equations are not referenced within the text
- The presented work is just a simulation work. The authors should justify the significance of presenting this simulation research.
- There is no clear scenario proposed as an infrastructure. A separate section with the proposed system architecture and explanations should be included.
- Provide the detailed simulation environment setting in case the performance of different approaches is evaluated by simulations.
Round 2
Reviewer 1 Report
The authors' proposed future work is intriguing and has potential for novelty and interest. However, I remain unconvinced that the presented work has sufficient novelty to qualify for publication in a journal. As previously noted, the manuscript could benefit from a more focused approach to the research question and a clear and concise presentation of the validation. To improve the manuscript, I suggest that the authors more directly address the novelty of their work by highlighting specific aspects that distinguish their approach from previous work in the field. This will help to motivate the audience and make a more compelling case for the significance of their research. In summary, the authors' proposed future work is promising and has potential for novelty and interest. However, the manuscript still requires further development to establish the significance of their research and to provide a more focused and concise presentation of their results.